# Evaluating Shortcut Utilization in Deep Learning Disease Classification through Counterfactual Analysis

**Vibujithan Vigneshwaran**[1,2]           VIBUJITHAN.VIGNESHWA@UCALGARY.CA

**Emma A.M. Stanley**[1,2]              EMMA.STANLEY@UCALGARY.CA

**Raissa Souza**[1,2]           RAISSA.SOUZADEANDRAD@UCALGARY.CA

**Erik Ohara**[1,2]                 ERIK.OHARA@UCALGARY.CA

**Matthias Wilms**[1,2,3,4,5]            MATTHIAS.WILMS@UCALGARY.CA

**Nils D. Forkert**[1,2,5]             NILS.FORKERT@UCALGARY.CA

[1] *Department of Radiology, University of Calgary, Calgary, AB, Canada*

[2] *Hotchkiss Brain Institute, University of Calgary, Calgary, AB, Canada*

[3] *Department of Pediatrics, University of Calgary, Calgary, AB, Canada*

[4] *Department of Community Health Sciences, University of Calgary, Calgary, AB, Canada*

[5] *Alberta Children's Hospital Research Institute, University of Calgary, Calgary, AB, Canada*

**Editors:** Accepted for publication at MIDL 2025

## Abstract

Although deep learning models can surpass human performance in many medical image analysis tasks, they remain vulnerable to algorithmic shortcuts, where spurious correlations in the data are exploited, which may lead to reduced trust in their predictions/classifications. This issue is especially concerning when models rely on protected attributes (*e.g.,* sex, race, or site) as shortcuts. Such shortcut reliance not only impairs their ability to generalize to unseen datasets but also raises fairness concerns, ultimately undermining their purpose for computer-aided diagnosis. Previous techniques for analyzing protected attributes, such as supervised prediction layer information tests, only highlight the presence of protected attributes in the feature space but do not confirm their role in solving the primary task. Determining the impact of protected attributes as shortcuts is particularly challenging, as it requires knowing how a model would perform without those attributes — a counterfactual scenario typically unattainable in real-world data. As a workaround, researchers have addressed the absence of counterfactuals by generating *synthetic* datasets with and without protected attributes. In this study, we propose a novel approach to evaluate *real-world* datasets and determine the extent to which each protected attribute is used as a shortcut in a classification task. Therefore, we define and train a causal generative model to produce causally-grounded counterfactuals, removing protected attributes from activations and allowing us to measure their impact on model performance. Employing T1-weighted MRI data from 9 sites (835 subjects: 426 with Parkinson's disease (PD) and 409 healthy), we demonstrate that counterfactually removing the 'site' attribute from the penultimate layer of a trained classification model reduced the AUROC for PD classification from 0.74 to 0.65, indicating a 9% performance improvement achieved by using 'site' as a shortcut. In contrast, counterfactually removing the 'sex' attribute had minimal impact on performance, with only a slight change of 0.004, indicating that 'sex' was not utilized as a shortcut by the classification model. The proposed method offers a robust framework for assessing shortcut utilization in medical image classification, paving the way for improved bias detection and mitigation in medical imaging tasks. The code for this work is available on GitHub.

**Keywords:** shortcuts, causality, counterfactual, Parkinson's disease, bias mitigation

## 1. Introduction

While deep learning models have surpassed human capabilities in many medical image analysis tasks, a significant concern are algorithmic shortcuts, where models learn spurious correlations in the training data, potentially resulting in biased predictions/classifications (Geirhos et al., 2020; Brown et al., 2023; Petersen et al., 2023). This issue becomes particularly problematic when models rely on protected attributes (*e.g.,* sex, race, or site) as shortcuts to make predictions. Such reliance hinders the ability of models to generalize to unseen datasets, ultimately undermining their intended purpose of computer-aided diagnosis and, thereby, reducing trust in model predictions. Thus, shortcuts — along with their causes, evaluation, and mitigation — have been studied across various subfields of medical image analysis, including aspects such as fairness, bias mitigation, and harmonization (Wang et al., 2024; Stanley et al., 2024; Souza et al., 2023; Vigneshwaran et al., 2024b). However, in any case, the crucial initial step is to identify the extent to which a model relies on shortcuts, a challenge that remains mostly unresolved.

Disparities in performance across protected attribute subgroups may indicate that the model is relying on shortcuts. However, it is not possible to conclude that a model is using these attributes as shortcuts solely based on performance disparities, as these differences may result from a combination of distribution shifts and shortcut learning (Castro et al., 2020). Researchers have explored the presence of protected attributes within disease classification models using methods like the supervised prediction layer information test (SPLIT), multitask learning (Glocker et al., 2023), and representation learning (Rane et al., 2024). SPLIT involves training the model for a primary task (*e.g.,* disease classification) and then retraining only the final layer for a secondary task (*e.g.,* race classification). For example, Souza et al. (2024b) demonstrated that even when attributes, such as site, scanner type, and sex, are not explicitly included, they can still be inferred from the penultimate layer of a deep learning disease classification model. However, the presence of protected attributes in the penultimate layer alone does not necessarily mean that these attributes are actually being used for the primary task. A good example of this is provided by Glocker et al. (2023), who demonstrated that even when a disease classification model's backbone is randomly initialized (*i.e.,* not trained), it can still infer information about racial identity in the penultimate layer. This is because current deep learning models have such high capacities that even random transformations can find some information about protected attributes in the model.

In order to determine if protected attributes are not only present in the model but are actually influencing the primary task, researchers have used visual feature space exploration techniques like principal component analysis (PCA) and t-SNE. These methods aim to explore the information encoded within a model, how it is distributed, and how it aligns with the primary task (Glocker et al., 2023). However, quantifying the extent to which protected attributes contribute as actual shortcuts may be challenging because it requires knowledge of how well the model *would* perform if it did not have access to those attributes. Such a *counterfactual* dataset is typically unattainable in real-world scenarios. As a preliminary step, Stanley et al. (2025) addressed this challenge by using a synthetic dataset to measure the degree of shortcut utilization in a classification model. Therefore, they generated *counterfactual* synthetic datasets, both with and without the presence of a protected attribute,

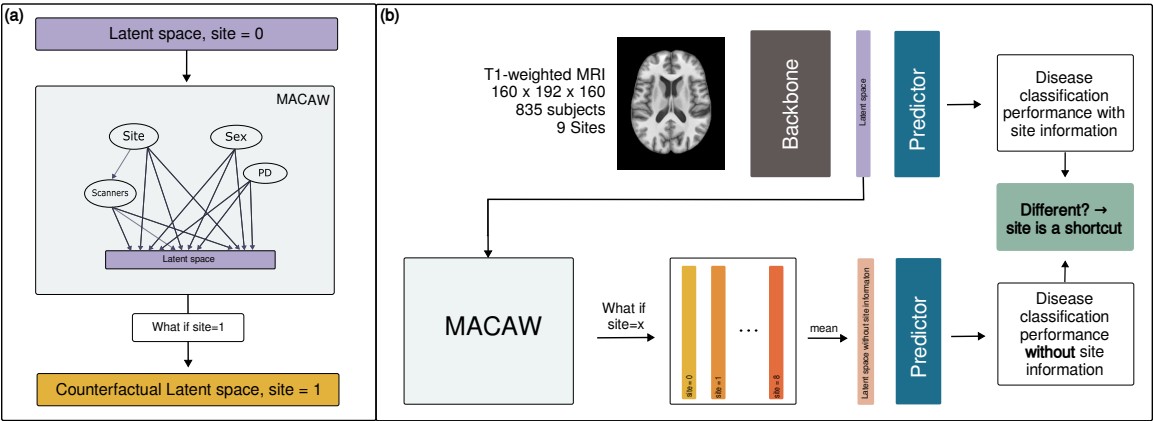

Figure 1: (a) A predefined causal graph was encoded into MACAW to generate counterfactual latents. (b) Counterfactual latents were generated for all sites and then averaged together to remove site-specific information. These site-removed latents were subsequently used for disease classification. If the classification accuracy differs from the standard setup (with site information), it suggests that site information is being used as a shortcut.

which enabled the quantification of the degree to which a shortcut was utilized in a deep learning classification task.

In this work, rather than relying on synthetic data that has limited usefulness in clinical applications, we propose using causally-grounded counterfactuals on *real-world* data. Specifically, we counterfactually remove latent representations of protected attributes to quantify shortcut utilization in classification models. In this context, counterfactuals represent hypothetical scenarios produced by querying causal generative models with questions such as "What would the latent representations look like if they did not contain any protected attributes?" and have recently been utilized in the field of algorithmic fairness (Kusner et al., 2018; Cornacchia et al., 2023). However, to the best of our knowledge, our work is the first to utilize counterfactuals to quantify the extent to which a deep learning image-based disease classification model relies on and makes use of shortcuts. It is important to note that our goal is not to develop a fair prediction model or to mitigate bias in the data. Instead, we aim to first determine whether certain protected attributes are used in deep learning disease prediction/classification models as shortcuts and then assess the extent of their contribution to the final prediction.

## 2. Methodology

As a representative case example, we first train a multi-site Parkinson's disease (PD) classification model and identify performance disparities across sex and site subgroups. First, quantitatively using SPLIT, and qualitatively using PCA, we demonstrate the presence of sex and site attributes in the penultimate layer. Next, we apply our proposed causal analysis-based technique to counterfactually remove these attributes and validate their re-

moval through updated SPLIT and PCA analyses. Finally, we evaluate the impact of these attributes on classification performance by computing differences in the model's accuracy.

## 2.1. Classification model

For PD classification, we used a modified SFCN model (Peng et al., 2021). In this modification, we replaced the original model's final $1 \times 1 \times 1$ convolution layer with a fully connected layer to map all channels to a single output neuron, which served as the logit. We refer to all layers except the final layer as the "backbone" and the final layer as the "prediction layer" throughout the remainder of this work.

## 2.2. Counterfactually removing protected attributes

Our approach to removing protected attributes from a trained classification model is based on asking the hypothetical question: "What would the activations in the penultimate layer look like if they did not contain any protected attributes?" This involves extracting activations from the penultimate layer and generating all possible counterfactual activations for a specific protected attribute. These counterfactual activations are then averaged to remove any information related to the protected attribute. For example, for the 'site' attribute, the extracted 64-dimensional activations from the penultimate layer for each image are referred to as $\mathbf{x}_i^{site}$, and the corresponding counterfactual activations are denoted as $\mathbf{x}_{i \to j}^{site}$, where the subscript $i$ represents the original site and the variable $j \in [1, n]$ indicates the counterfactual site, and $n$ is the number of sites. The activation with the site attribute removed for each image is denoted as $\mathbf{x}^{-site}$ and is calculated as follows:

$$\mathbf{x}^{-site} = \frac{1}{n} \sum_{j=1}^{n} \mathbf{x}_{i \to j}^{site} \tag{1}$$

The causal analysis used in the background of the proposed method to remove potential shortcuts starts with a directed acyclic graph (DAG), which represents causal relationships between variables. In a DAG, each node corresponds to a variable, and each directed edge indicates a causal influence. In this work, the MACAW framework (Vigneshwaran et al., 2024a) was used as the causal generative model (Fig. 1).

### 2.2.1. MACAW

This causal generative model uses normalizing flows to transform complex probability distribution into Gaussian distribution, which can then be used to generate new data thorough sampling or generate counterfactuals though causal interventions. The causal domain knowledge is encoded to the normalizing flow by masking connections to preserve the causal dependencies between parent and child nodes. Once the normalizing flow is masked, the model is optimized through maximum likelihood estimation. After training the model, counterfactuals can be easily generated to explore hypothesized alternate scenarios. Following the terminology of (Pearl, 2012), counterfactual generation consists of three steps: abduction, action, and prediction. In MACAW, abduction involves mapping the observed real sample into the transformed Gaussian distribution, action modifies a causal variable

(*e.g.*, changing 'site 0' to 'site 1'), and 'prediction' transforms the modified latent representation back into the original space. The MACAW architecture diagram is presented in the Appendix for reference. For a detailed discussion of the theoretical and implementation of the MACAW model, readers are referred to (Vigneshwaran et al., 2024a).

### 2.3. The supervised prediction layer information test (SPLIT)

SPLIT was proposed by Glocker et al. (2023) to confirm the presence of protected attributes in the penultimate layer. Therefore, the SFCN model described above was trained until convergence to classify PD and healthy subjects. Then, all backbone parameters were frozen, and the prediction layer was replaced with a new one. This new prediction layer was trained specifically to classify protected attributes, including site and sex in this work, learning a new set of weights assigned to the features in the penultimate layer. A model (prediction layer) was trained for sex classification, while separate models were trained for each site using a one-vs-rest approach. Each model was trained until convergence, and the model parameters with the best validation loss were selected. The performance of the new prediction layer was then evaluated on test data.

### 2.4. Unsupervised feature space exploration with PCA

This analysis projects the model's activation space onto a lower-dimensional PCA space and qualitatively evaluates whether the PCA modes for PD classification and those for the protected attributes form visible groupings in scatter plots. If such groupings are observed, it suggests that the protected attributes are encoded in the latent space, and possibly being used as shortcuts. Therefore, features (activations) from the penultimate layer were first extracted by passing the test data through the backbone of the SFCN model, resulting in a 64-dimensional feature vector for each test image. Next, the feature space was decomposed into principal components using PCA. The first and second principal components were marginalized and visualized for PD, site, and sex to qualitatively identify patterns.

## 3. Experiments

### 3.1. Data

We utilized brain imaging data from nine sites, including 835 subjects (426 with PD and 409 healthy) to develop and evaluate our shortcut utlization evaluation method. The studies included were the BioCog (Clinical, Magnetic Resonance, and Genetic Biomarkers of Cognitive Decline and Dementia in Parkinson's Disease) (Acharya et al., 2007), C-BIG (Montreal Neurological Institute's Open Science Clinical Biological Imaging and Genetic Repository) (Das et al., 2022), Neurocon (Badea et al., 2017), Tao Wu (Badea et al., 2017), OpenNeuro Japan(Yoneyama et al., 2018), PD MCI Calgary (Lang et al., 2019), and Hamburg (Talai et al., 2021). For this work, we included subjects with T1-weighted magnetic resonance imaging (MRI) datasets and known ground truth labels. All studies received ethics approval from their respective local ethics boards, and written informed consent was obtained from all participants in compliance with the Declaration of Helsinki. In the preprocessing step (Camacho et al., 2023), the T1-weighted images underwent skull-stripping using HD-BET (Isensee et al., 2019), resampling to 1 mm isotropic resolution with linear interpolation, and

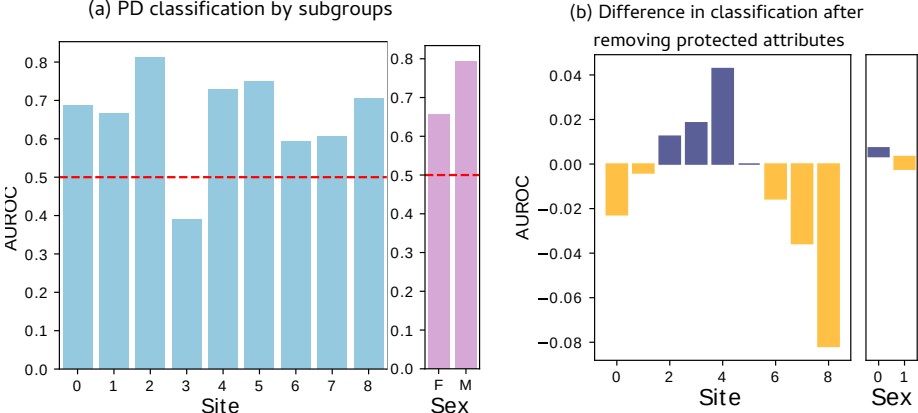

Figure 2: (a) The model's performance for Parkinson's disease (PD) classification was analyzed across different subgroups, revealing significant variations in AUROC scores. (b) Site- and sex-wise AUROC changes after removing the corresponding protected attributes. The red line corresponds to chance level performance.

bias field correction (Tustison et al., 2010). Each T1-weighted MRI data was then affinely registered to the PD25-T1-MPRAGE- 1 mm atlas (Xiao et al., 2017) and center-cropped to $160 \times 192 \times 160$ voxels. The data was stratified by site and sex and split into training (50%), validation (10%), and testing (40%) sets. The following integer identifiers were assigned to the sites for reference: 'PD MCI Calgary: 0, UKBB: 1, Hamburg: 2, C-BIG: 3, Neurocon: 4, OpenNeuro Japan: 5, PD MCI PLS: 6, Taowu: 7, BioCog: 8' (Appendix: Fig. 6).

### 3.2. PD classification

The classification model was trained until convergence, and the model with the best validation loss was selected. The following hyperparameters were used to achieve the best validation loss: number of channels = [28, 58, 128, 256, 256, 64], learning rate = $1 \times 10^{-4}$, and weight decay = $1 \times 10^{-5}$. PyTorch was used for the implementation of the model, which was trained on an NVIDIA RTX 3090 GPU. After training, the model was evaluated on the test set, achieving an overall area under the receiver operating characteristics (AUROC) of 0.74. This performance is comparable to other multisite PD classification studies (Camacho et al., 2023). Subsequently, we analyzed the performance of the model across individual sites and sexes. As illustrated in Fig. 2(a), different sites exhibited varying performance, with site 2 achieving an AUROC of 0.81, while site 3 had an AUROC of 0.38. Additionally, a disparity in performance between male (0.65) and female (0.79) participants was observed. These discrepancies might be related to algorithmic shortcutting or underlying distribution shifts.

### 3.3. Presence of protected attributes

When performing SPLIT on the data for each site to determine the presence of site-specific information in the penultimate layer, certain sites (sites 6, and 8) were easily distinguishable

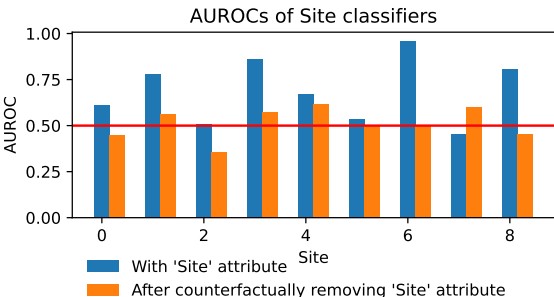

Figure 3: The AUROC for site classification was compared before and after the counterfactual removal of the site attribute.

from others, achieving AUROC values close to 0.9. However, some sites (sites 2, 5, and 7) had AUROC values close to chance level (Fig. 3). Additionally, sex classification using SPLIT on the penultimate layer achieved an AUROC of 0.62, suggesting the presence of protected attributes to some extent.

Next, we analyzed the activation space of the penultimate layer using PCA. The results revealed visible groupings of sites in the scatter plots, and the distributions of certain sites aligned more closely with healthy subjects, while others aligned with PD subjects (Fig. 4(A)). However, no distinct alignment patterns were observed when comparing sex distributions (Appendix: Fig. 8). These findings suggest that site information may serve as a shortcut to some extent in the trained PD deep-learning classification model.

### 3.4. Removing protected attributes counterfactually

We first predefined a DAG to incorporate domain knowledge into the MACAW model. The causal graph used in this work is shown in Fig. 1(a), where sex, site, scanner, and PD status are set as causal variables influencing the activations. The prior distributions for each causal variable were estimated from the training data. For sex and PD, a Bernoulli distribution was used, while site values were one-hot encoded and modeled with a One-Hot categorical distribution as the prior. The model was trained until convergence, and the version with the best validation loss was selected. The following hyperparameters were used to achieve the best validation loss: hidden layer multipliers = [4,6,4], learning rate = $1 \times 10^{-4}$, weight decay = $1 \times 10^{-5}$, and number of MACAW layers = 6.

We then counterfactually removed site and sex information from the activation layers in separate experiments (*i.e.,* one for site removal and another for sex removal). After counterfactually removing these attributes from the penultimate layer, we evaluated their presence using SPLIT (Fig. 3). The AUROCs for all sites dropped to approximately 0.5 (to chance level), while the sex classification AUROC decreased from 0.62 to 0.47. These results confirm that the counterfactual method successfully removed site and sex information from the penultimate layer.

Furthermore, we analyzed the PCA modes after the removal of the site and sex attributes. As shown in Figure 4(B), the site distributions became more homogenous, with

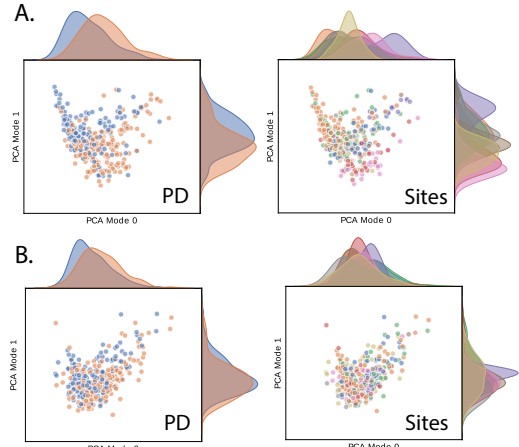

Figure 4: A. PCA modes 0 and 1 of the activation space were visualized based on PD labels and site labels. B. the visualization was then repeated after counterfactually removing site attribute from the activations.

no groupings or alignment observed between specific sites and PD classes. Similarly, after the removal of the sex attribute, the male and female distributions overlapped more strongly, indicating the successful removal of the sex attribute. However, there was minimal change in the PD distributions after removing the sex attribute, suggesting that sex was not utilized as a shortcut in the classification task. (Appendix: Fig. 8).

### 3.5. Classification performance after removing protected attributes

We hypothesized that if the model utilized protected attributes as shortcuts, removing these attributes would result in a change in model performance, indicating that performance disparities were caused by shortcuts. After removing the site attribute, the PD classification AUROC decreased from 0.74 to 0.65, demonstrating a 0.09 performance drop that may be attributable to the use of the site as a shortcut. In contrast, removing the sex attribute resulted in a negligible performance change (0.004), indicating that sex information was minimally present in the penultimate layer and not or only to a very small extent used as a shortcut for PD classification. Consequently, the removal of sex information had little impact on the model's performance.

We further analyzed site- and sex-wise shortcut utilization within subgroups. Fig. 2(b) illustrates the AUROC differences observed after removing protected attributes. Notably, removing the site attribute led to significant performance reductions for the data from certain sites (*e.g.,* sites 7 and 8), while removing the sex attribute caused minimal changes. Interestingly, the data from some sites and the female population showed slight AUROC improvements after attribute removal, which may suggest that the model is shifting toward better generalization. However, a more comprehensive analysis, including comparisons of sample sizes and PD distribution across subgroups, is required to validate this hypothesis.

## 4. Conclusion

We demonstrated in this work that causally-grounded counterfactuals can be used for a quantification of the the extent to which protected attributes are used as shortcuts in classification tasks. Our findings revealed that counterfactually removing the site attribute from the penultimate layer resulted in a 0.09 AUROC reduction in PD classification, indicating that site-related information was used as a shortcut in the initial classification model. Conversely, removing the sex attribute had minimal impact on classification performance, suggesting that sex was not utilized as a shortcut.

Previous research has utilized adversarial multitask learning (Brown et al., 2023; Souza et al., 2024a) and distributionally robust optimization (DRO) (Sagawa et al., 2019) techniques to remove or disentangle biases in the model performance. However, we believe that directly comparing our approach to these methods would not be a one-to-one. The key difference lies in the objective: bias-mitigation techniques aim to remove bias while allowing the model to adjust its representation of the disease to achieve fairer and more accurate results. In contrast, our approach focuses on the protected attribute to observe the extent to which the model had been relying on them for disease classification. This step, similar to SPLIT, is critical for determining whether the attributes are acting as shortcuts before/during applying any bias mitigation techniques. Additionally, the proposed method can be used to evaluate the effectiveness of bias mitigation approaches in eliminating shortcuts from the dataset. For example, we optimized an SFCN model using Group-DRO and then removed site information from the penultimate layer. After removal, AUROC dropped from 80% to 66%, suggesting that while Group-DRO improves generalization, it still relies on site information. Detailed results are in Appendix D.

A key limitation of this work is the lack of external validation for the MACAW model. One could argue that the accuracy drop is due to MACAW distorting the latent space rather than removing shortcuts. Typically, two key metrics are used to validate counterfactuals: effectiveness (whether the counterfactual generation achieves the intended outcome) and amplification (whether the counterfactual generation does not alter anything undesired). Although we did not use an external classifier (Monteiro et al., 2023), we noted similar results in our sex-removal experiment. In this experiment, after counterfactually removing the sex attribute, we found that the sex classification dropped to chance level (indicating effectiveness), while the PD accuracy remained unchanged (indicating no unwanted amplification). This result implicitly indicates that the MACAW counterfactual generation is valid. If the counterfactual generation were noisy, we would expect to see significant drops in both sex and PD classification AUROC. However, a more controlled experimental scenario, using a synthetic dataset (Stanley et al., 2024), would provide a clearer proof of concept and allow for a more thorough evaluation of the counterfactual approach.

Another limitation of this work lies in the predefined causal graph used. We selected a graph where the PD variable is not influenced by other factors, enabling straightforward disentanglement of PD from protected attributes. However, alternative causal graph structures are also plausible. Having said that, we believe that this method can still be used as a tool for technical research to explore biases and strategies for reducing them. Given the MACAW model's capability to handle complex causal structures, this technique is highly adaptable to other datasets/diseases, and deep learning models.

## Acknowledgments

This work was supported by the Canadian Neuroanalytics scholar (CNS) program, the Canada Research Chairs program, the River Fund at Calgary Foundation, and the NSERC Discovery Grant program.

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

## Appendix A. MACAW architecture

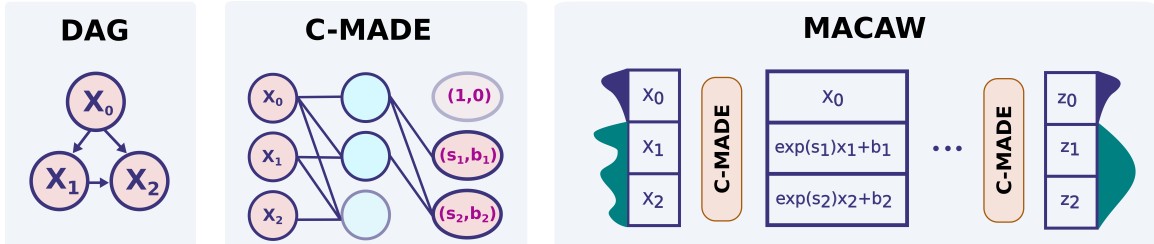

Figure 5: A causal DAG (left) and its respective C-MADE network (center). The MACAW architecture (right) consists of multiple C-MADE networks connected in a series, thereby forming a normalizing flow.

## Appendix B. Data distribution

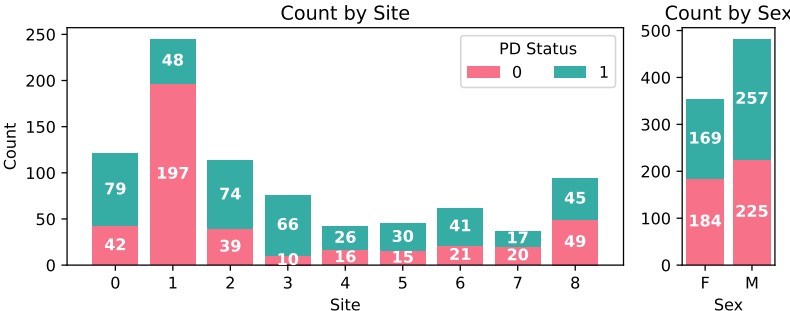

Figure 6: Total data distribution by protected attributes - site and sex.

| Site | | 0 | | | 1 | | | 2 | | | 3 | | | 4 | | | 5 | | | 6 | | | 7 | | | 8 | | |
|---|---|---|---|---|---|---|---|---|---|---|---|---|---|---|---|---|---|---|---|---|---|---|---|---|---|---|---|---|
| Source | | Tr | V | Te | Tr | V | Te | Tr | V | Te | Tr | V | Te | Tr | V | Te | Tr | Va | Te | Tr | V | Te | Tr | V | Te | Tr | V | Te |
| Category | Value | | | | | | | | | | | | | | | | | | | | | | | | | | | |
| PD | 0 | 21 | 4 | 17 | 98 | 20 | 79 | 20 | 3 | 16 | 4 | 1 | 4 | 8 | 1 | 7 | 7 | 2 | 6 | 11 | 2 | 8 | 10 | 2 | 8 | 25 | 5 | 19 |
| | 1 | 40 | 8 | 31 | 24 | 5 | 19 | 37 | 7 | 30 | 32 | 7 | 27 | 13 | 3 | 10 | 15 | 3 | 12 | 21 | 4 | 16 | 8 | 2 | 7 | 22 | 5 | 18 |
| Sex | 0 | 24 | 5 | 19 | 49 | 10 | 39 | 19 | 3 | 15 | 19 | 4 | 16 | 11 | 2 | 9 | 12 | 3 | 10 | 14 | 2 | 10 | 8 | 2 | 7 | 21 | 4 | 16 |
| | 1 | 37 | 7 | 29 | 73 | 15 | 59 | 38 | 7 | 31 | 17 | 4 | 15 | 10 | 2 | 8 | 10 | 2 | 8 | 18 | 4 | 14 | 10 | 2 | 8 | 26 | 6 | 21 |

## Appendix C. Sex PCA distribution after counterfactuals

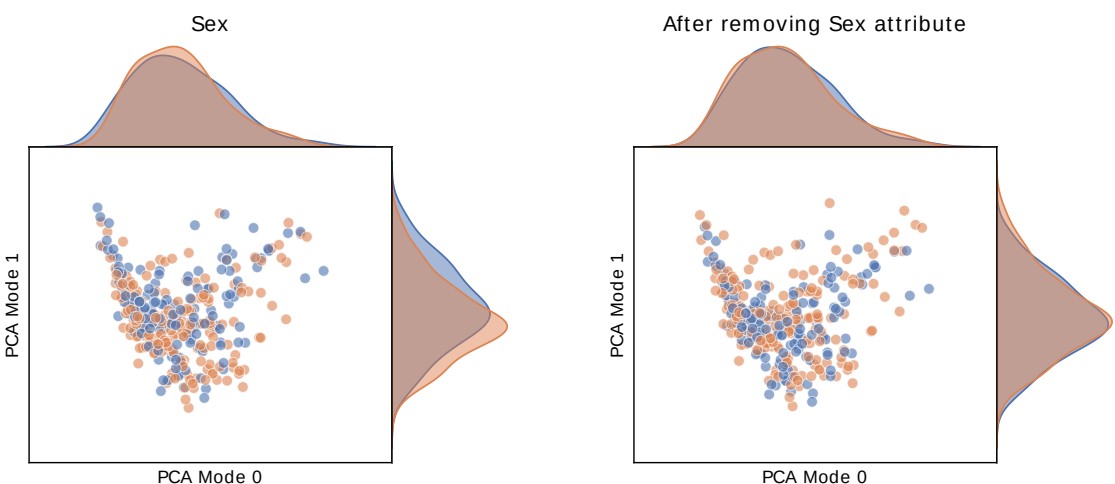

Figure 7: PCA modes 1 and 2 of the activation space were visualized based on sex, and after counterfactually removing sex attribute from the activations.

## Appendix D. GroupDRO training

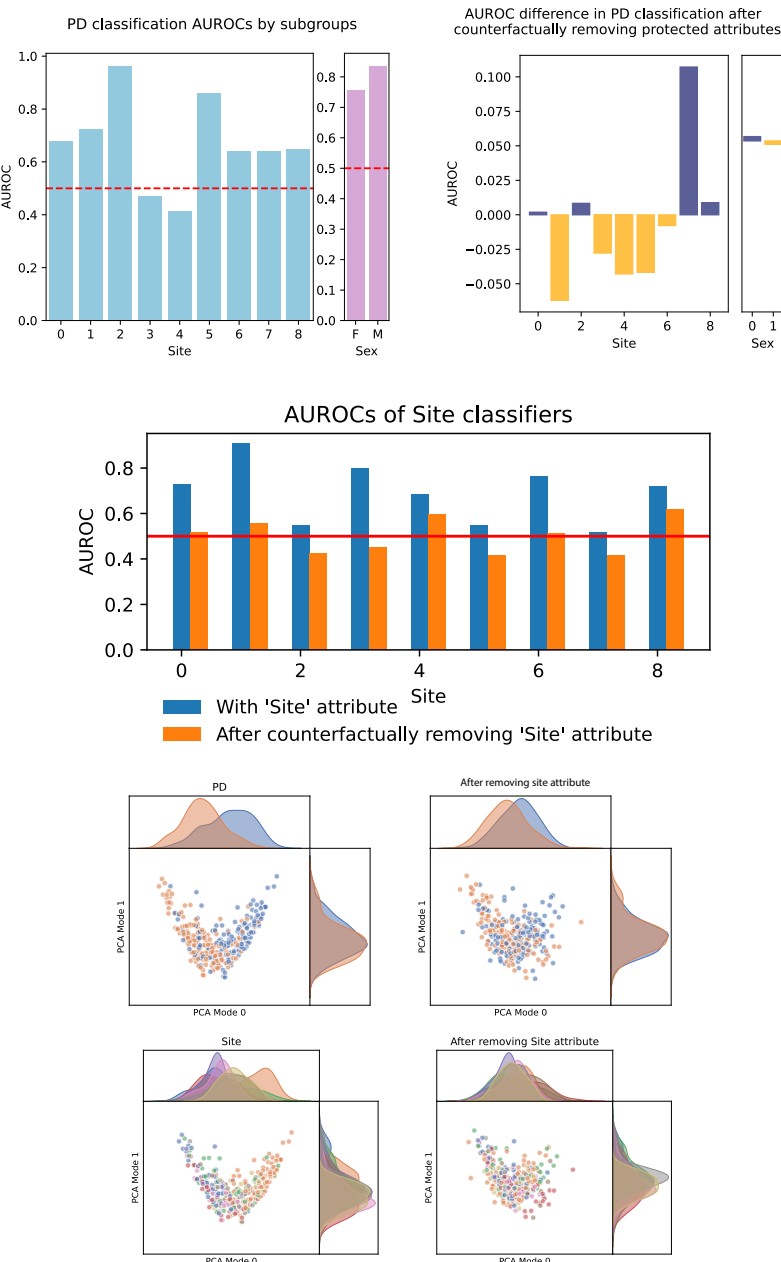

Figure 8: The results for the SFCN model using GroupDRO optimization are presented. In this approach, instead of minimizing the average loss across the training samples, the optimization assigns greater weight to the worst-performing site.

