# OpenReview forum: "Evaluating Shortcut Utilization in Deep Learning Disease Classification through Counterfactual Analysis"
_MIDL.io/2025/Conference — MIDL 2025 Oral_

### Official Review · Reviewer_FFqq · 2025-02-15

**Confidence:** 4
**Preliminary Rating:** 3
**Recommendation:** Poster
**Final Rating:** 5

**Summary:**

The paper proposes to use counterfactual images to analyze the performance of shortcuts in disease classification models. Results demonstrate that this analysis could be effective in discovering the effect of shortcuts on classification performance.

**Strengths:**

* The introduction is really well written, with clear motivation and a good literature review.
* The code, including scripts to reproduce the images in the paper, is publicly available. This is commendable.
* Good presentation of the results.

**Weaknesses:**

* **Missing details about MACAW**: One big concern about the paper is the lack of explanation regarding the utilized MACAW method. Although the paper’s main focus is not on MACAW but rather on its application for shortcut learning, it becomes difficult to understand the methodology without a sufficient explanation of MACAW.
* **Missing method figure**: It would be easier if there was a small method figure, which describes all the main parts of the proposed method. For example, the MACAW framework, its application for counterfactual scenarios in classification, and its utility in the analysis of shortcut learning. If the authors face space constraints, they can try to shorten the Abstract.
* **Repetition**: Some minor things, like the dataset details, are repeated in two different sections (Sec 2.1 and 3.1). It might be a good idea to talk about the general classification network and classification paradigm in section 2.1 and keep experimental details for sec 3.1. Same for the end of Sec 2.2, implementation details mentioned at the end could be moved to Sec 3.1.
* **Rationale for ONEvsALL site classification**: Can authors comment on why the OvR approach was utilized for site classification in SPLIT? Was this approach taken based on recommendations from the authors of SPLIT?
* **More insights in Figure 1**: It might be a good idea to provide more insights into AUROC performance for PD classification vs site classification. For example, a quick observation tells us that sides 3 and 6 have the highest AUROC for classification and the lowest AUROC for PD classification. Is there any rationale behind this?
* **Merging of Figures 1 and 3 (b)**: It would be good if the authors merge Figures 1 and 3 (b). It would make it easier to compare performance before and after site removal.
* **Other comparisons**: It might be a good idea to analyze the effect of site removal on the performance of classification networks trained using alternative methods like adversarial training or Groiup-DRO. As the primary focus of the work is on analysis rather than new methods for bias mitigation, it might be a good idea to understand these effects with different classification training methods. Similarly, have authors tried any other methods for counterfactual image generation? For example. rather than employing MACAW with normalizing flows, they could utilize HVAE [1] or Diffusion models [2]. It might not change the underlying analysis performed in this paper, but considering that both of them have shown good image quality [3], it might show a better analysis.

[1] Ribeiro, F.D.S., Xia, T., Monteiro, M., Pawlowski, N. and Glocker, B., 2023. High fidelity image counterfactuals with probabilistic causal models. arXiv preprint arXiv:2306.15764.

[2] Sanchez, P. and Tsaftaris, S.A., 2022. Diffusion causal models for counterfactual estimation. arXiv preprint arXiv:2202.10166.

[3] Melistas, T., Spyrou, N., Gkouti, N., Sanchez, P., Vlontzos, A., Panagakis, Y., Papanastasiou, G. and Tsaftaris, S.A., 2024. Benchmarking counterfactual image generation. arXiv preprint arXiv:2403.20287.

**Detailed Comments:**

* Although the literature review is good, it is missing a couple of references related to fairness and causality. It might be a good idea to include them, too. [1,2,3]
* It might be a good idea to move figures to the same page (and also in the same order) as they are referred to in the paper. For example, Figure-3 (a) is referred to on page 4, while it is given on Page 3. Same for Figure 4. Also, Figure 3 and Figure 4 are referred to in the paper before Figure 1 or 2. Changing their order would make the paper much easier to navigate and read.

[1] Petersen, E., Holm, S., Ganz, M. and Feragen, A., 2023. The path toward equal performance in medical machine learning. Patterns, 4(7).

[2] Wang, R., Kuo, P.C., Chen, L.C., Seastedt, K.P., Gichoya, J.W. and Celi, L.A., 2024. Drop the shortcuts: image augmentation improves fairness and decreases AI detection of race and other demographics from medical images. EBioMedicine, 102.

[3] Rane, R.P., Kim, J., Umesha, A., Stark, D., Schulz, M.A. and Ritter, K., 2024, October. DeepRepViz: Identifying Potential Confounders in Deep Learning Model Predictions. In International Conference on Medical Image Computing and Computer-Assisted Intervention (pp. 186-196). Cham: Springer Nature Switzerland.

**Justification Of The Final Rating:**

The authors have clarified all the points raised during the initial review. Their results are really good. I think this work will serve as a good way of verifying the extent to which protected attributes are used in a robustness/fairness literature for image classification. Considering this, I recommend a strong accept, with oral presentation and MELBA extension.

**Justification Of The Preliminary Rating:**

Overall, the paper tackles an important problem and shows promising results. However, the methodology and experiment sections of the paper require rewriting. I have given detailed comments above. Considering that I am going with a borderline rating. I would be happy to increase it if the authors can address some of them during the rebuttal period.

**Questions To Address In The Rebuttal:**

Please try to address the points addressed in the weakness section (except the second half of the last one, which might not be feasible due to time).

**Special Issue:**

No

---

> ### Author Response · Authors · 2025-03-08
> **Point-by-point response to reviewer FFqq**
>
> We appreciate the reviewer's detailed feedback and for highlighting areas for improvement in our paper. Below, we have addressed each concern on a point-by-point basis.
>
> **Missing Details About MACAW:**
> We have added a section to the paper detailing the core aspects of the MACAW model. Additionally, a diagram summarizing the architecture of MACAW has been included in the Appendix to better illustrate the technique.
>
> **Missing Method Figure:**
> Thanks for the suggestion. We have now added a figure that visually represents the entire methodological process/workflow, similar to a graphical abstract.
>
> **Repetition:**
> We have removed the repeated dataset details. Also, we separated the description of the general architecture from the implementation details by moving the latter to the respective experiment sections.
>
> **Rationale for ONEvsALL Site Classification:**
> Yes, that’s correct. The decision to use the OvR approach was based on the SPLIT paper by Glocker et al., where they employed OvR for race classification. We followed the same methodology. However, we do not believe this to be a limiting factor of our analysis but rather see it as a possible design choice.
>
> **More Insights in Figure 1:**
> Unfortunately, we cannot provide a compelling reason why sites 3 and 6 show higher accuracy for site classification but lower accuracy for PD classification. One possible explanation might be that site 3 has the most imbalanced distribution among all sites. Another factor could be differences in intensity distributions.
>
> **Merging of Figures 1 and 3(b) :**
> We have rearranged the figures to first present the PD classification accuracy and its changes after applying the counterfactual approach, followed by a separate figure for the site-wise SPLIT.
>
> ### **Other Comparisons:**
> **It might be a good idea to analyze the effect of site removal on the performance of classification networks trained using alternative methods like adversarial training or Group-DRO**.
>
> That’s an interesting suggestion. We trained the SFCN model using Group-DRO (with site as the sensitive attribute, gamma = 1, and step size = 0.01), which improved the overall AUROC from 74% to 80%. Additionally, subgroup accuracy showed some generalization.
>
> However, after removing the site attribute using counterfactuals, the AUROC dropped to 66%, similar to standard training. This suggests that while robust training improves generalization, the model still heavily relies on site information. A more controlled setup is needed to confirm this. Thus, we have briefly mentioned this in the discussion and added the results to the Appendix. Additionally, the codebase has been updated to include the Group-DRO training.
>
> **Other Methods for Counterfactual Image Generation**
> HVAE and Diff-SCM are indeed more effective for generating image counterfactuals. However, our work here operates in the activation space of a neural network rather than directly on images; thus, a straightforward approach like MACAW, which embeds the causal DAG into a single normalizing flow, is easier to implement for this specific case.
>
> Additionally, we have included the references suggested by the reviewer and reorganized the figures to improve readability. Thank you for these excellent suggestions.

---

> > ### Comment · Reviewer_FFqq · 2025-03-08
> > **Response to Author Rebuttal**
> >
> > * I thank the authors for incorporating all the major and minor points in the revised manuscript.
> > * **Group-DRO**: Overall, I am quite happy with the rebuttal provided by the authors. I am particularly impressed with the observation made regarding models trained using GroupDRO. The authors' analysis shows that GroupDRO models are not "removing" the dependence on the protected attributes but only utilizing it more than ERM-based models. If possible, I suggest the authors include this in the main paper results.
> > * **Validation of MACAW**: As suggested by the authors in the discussion section, validation of MACAW is necessary. I agree with the authors that SPLIT analysis provides a good proxy for this. However, I think a more direct approach could be to plot density plots of PCA modes for "original" and "counterfactual" latent space. For example. authors could take the "original" latent space of site-1 images and the "counterfactual" latent space of converting all other sites to site-1. This would verify that at least both "original" and "counterfactual" latent spaces are following the same distributions.

---

> > ### Author Response · Authors · 2025-03-11
> > **Response to Reviewer FFqq**
> >
> > We thank the reviewer for their valuable feedback on the manuscript.
> >
> > **Group-DRO**
> >
> > We found this topic intriguing as well. It appears that site information, when used in DRO loss computation, leaks into the model, helping it generalize better. Conceptually, this result makes sense if we view Group-DRO as positioned between two extremes: training a single model using ERM and training separate models for each site. We plan to combine the results of ERM training with Group-DRO in the results section.
> >
> > **Validation of MACAW**
> >
> > Thank you for the suggestions. We have generated the relevant figures, which show that the counterfactual distributions from sites tend to align with the base site's distribution. We will include these results in the appendix. In the meantime, they can be viewed in the GitHub repository under https://github.com/vibujithan/shortcut-analysis/blob/main/PD/7_remove_attributes.ipynb

---

> > > ### Comment · Reviewer_FFqq · 2025-03-11
> > > **Final Response**
> > >
> > > Thank you for incorporating these changes. These results are quite impressive and show that MACAW is generating good counterfactual latent space.
> > >
> > > I will change my rating to strong accept.
> > >
> > > Looking forward to the presentation at MIDL. :)

---

### Official Review · Reviewer_fBp8 · 2025-02-19

**Confidence:** 5
**Preliminary Rating:** 2

**Summary:**

This study applies a method to remove the spurious correlations of site and sex information from a model that predicts Parkinson’s disease from MRI images.

**Strengths:**

The task is very relevant to building safe and effective medical imaging analysis tools.
................................................................................................................

**Weaknesses:**

The proposed methods for generating counterfactual examples and removing the spurious signal are not clear to me.
................................................................................................................

**Detailed Comments:**

See next section

**Justification Of The Preliminary Rating:**

This work potentially contains interesting results however the method is too opaque to replicate. The results are also not tested for significance which hinders the impact that this work could have.
................................................................................................................

**Questions To Address In The Rebuttal:**

The proposed method is not clear to me. Mainly how the counterfactuals are generated and how the spurious information is removed. I think I follow that you are taking the average of these representations but I don't follow why this would work. I think scaling of the samples in the mean could be important here if some counterfactuals have too much signal removed and overshoot. But I also don't see a description of how the counterfactuals are generated to clear this up.

Also, without significant testing it is not possible to conclude that this approach offers an improvement. Please compute a p-value for the claimed improvement, specifically for those in 3.4 and 3.5.

It would also be useful to include an existing baseline approach (such as GroupDRO) for comparison.

The font in figure 2 is too small.

**Special Issue:**

No

---

> ### Author Response · Authors · 2025-03-08
> **Point-by-point response to reviewer fBp8**
>
> We thank the reviewer for the constructive feedback. We have addressed the concerns on a point-by-point basis below:
>
> **The proposed method is not clear to me:**
>
> To improve the clarity and readability of the manuscript, we have made the following revisions:
>
> - We have added a graphical abstract at the beginning to provide an overview of the entire process from start to finish in a single figure.
> - A new section outlining how the MACAW counterfactual generation works has been included.
> - We have moved the implementation details to the experiments section, where the methodology now focuses on the theory and architectural details.
> - We have merged Figures 1(a) and 3(b) for better readability and reorganized the figure order to enhance the overall flow of the paper.
> - The font sizes in figures have been increased for better visibility.
>
> **Also, without significant testing it is not possible to conclude that this approach offers an improvement.**
>
> We would first like to clarify that our approach does not aim to improve bias mitigation or similar aspects of fairness considerations but is essentially a new tool to precisely analyze shortcut learning in the latent space of neural networks.
> However, one way to assess significance of the analyses possible with the proposed technique is through k-fold cross-validation to obtain a p-value. But given that some sites have a limited number of samples, performing cross-validation with a stratified split of sex, site, and PD status would become infeasible. Typically, effectiveness and amplification metrics are used to determine if the counterfactuals are generated as intended and whether any unintended changes were introduced. This is inherently demonstrated in our sex-removal experiment, where the sex classification dropped to chance level after removal (effectiveness), but the PD accuracy remained unchanged (amplification). This result not only shows that sex is not being used as a shortcut but also implicitly indicates that the counterfactual generation is working as expected. If the counterfactual generation were noisy, we would expect significant drops in both sex and PD classification AUROC. We clarified this in the manuscript.
>
> **It would also be useful to include an existing baseline approach (such as GroupDRO) for comparison.**
>
> As indicated above, the primary focus of this paper is not bias mitigation; rather, our method aims to examine whether protected attributes are being used as shortcuts. The conceptually important distinction lies in the fact that bias mitigation techniques work to remove bias while allowing the model to achieve fairer results across different subgroups. For example, Distributionally Robust Optimization (DRO) minimizes the worst-case training loss over a set of predefined groups. In contrast, our approach removes the protected attribute to observe how the model's performance is affected if this potential shortcut is unavailable. This step, similar to SPLIT, is essential for determining whether the attributes are being used as shortcuts before implementing any bias mitigation techniques. However, the proposed technique allows us to evaluate the effectiveness of bias mitigation methods, such as GroupDRO or adversarial training, in removing shortcuts from this dataset. To assess the extent to which shortcuts are utilized, we trained a network using the GroupDRO method and compared it to the standard setup. The results have now been added to the discussion.

---

### Official Review · Reviewer_hxHa · 2025-02-21

**Confidence:** 4
**Preliminary Rating:** 4
**Recommendation:** Poster
**Final Rating:** 4

**Summary:**

This paper investigates the impact of specifically annotated shortcuts in a classification task by generating an “unbiased” representation for each sample. The approach involves creating counterfactual activations across all values of a particular protected attribute and averaging them to form the unbiased representation, which is then compared to the original “biased” one. They investigate two protected attributes (sex, and site) for the Parkinson's disease classification task. Although the procedure effectively reduces the presence of the protected attribute in the representation (debiasing), it also adversely affects disease classification performance.

**Strengths:**

+ The paper addresses a challenging and clinically significant problem, making it highly valuable to the medical community.
+ The manuscript is well written and accessible.
+ The innovative use of counterfactuals to generate unbiased representations and measure the effect of shortcuts.
+ Evaluates the effects of two protected attributes: sex and site (source institution).

**Weaknesses:**

- The main weakness is the absence of experiments to validate that the counterfactual generation preserves diagnostic information. Without such evaluations, it remains unclear whether the drop in performance is solely due to the removal of spurious protected attribute correlations or if it reflects a deterioration of the overall representation. For future work (not the rebuttal), this paper might bring some inspiration: Monteiro, Miguel, et al. "Measuring axiomatic soundness of counterfactual image models.", ICLR 2023.
- A more controlled experimental scenario would facilitate a clearer proof of concept and enable a more thorough evaluation of the counterfactual approach.

**Detailed Comments:**

Out of curiosity: Would a comparison to adversarial methods (that could also be used to create "unbiased" representations of features make sense?

**Justification Of The Final Rating:**

I was satisfied with the paper before the rebuttal and appreciated the authors' commentary. The paper has potential, and I think the community would appreciate an extension with counterfactual evaluation and more controlled setups.
Still, I recommend acceptance of the paper in its current state.

**Justification Of The Preliminary Rating:**

It is an interesting paper, that tackles an important evaluation problem of modern medical AI. However, the paper lacks experiments to validate the quality of the generated counterfactuals. This could be done through controllable synthetic experiments, or employing metrics from the literature, as the paper I suggested earlier. Without proper counterfactuals, the measurement of the impact of the spurious correlation might be too noisy.

**Questions To Address In The Rebuttal:**

None of the weakness I detected could be solved in the rebuttal.

---

> ### Author Response · Authors · 2025-03-08
> **Response to Reviewer hxHa**
>
> We thank the reviewer for the constructive feedback.
>
> **The reviewer is correct in noting that external validation is important for assessing the counterfactual results.**
>
> Typically, two key metrics are used to validate counterfactuals: effectiveness (whether the counterfactual generation achieves the intended outcome) and amplification (whether the counterfactual generation does not alter anything undesired). Although we did not use an external pseudo-oracle (such as a classifier model), we noted similar results in our sex-removal experiment. In this experiment, after counterfactually removing the sex attribute, we found that the sex classification dropped to chance level (indicating effectiveness of the intervention), while the PD accuracy remained unchanged (indicating no unwanted amplification). This result suggests that sex is not being used as a shortcut and implicitly indicates that the counterfactual generation is working as expected. If the counterfactual generation process was noisy/ineffective, we would expect to see significant drops in both sex and PD classification AUROC. However, this approach is only applicable when protected attributes are not used as shortcuts. As the reviewer pointed out, a more controlled experimental scenario, using a synthetic dataset such as SimBA (https://github.com/estanley16/SimBA), could provide a clearer proof in the future. We clarified this in the manuscript.
>
> **Out of curiosity: Would a comparison to adversarial methods (that could also be used to create "unbiased" representations of features make sense?**
>
> We believe that comparing our approach to a bias-mitigation technique would not be a relevant one-to-one comparison. The key distinction is that bias-mitigation techniques aim to remove bias while allowing the model to adjust its representation of the disease to achieve fairer and more accurate results. In contrast, our approach simply removes the protected attribute to observe how the model's performance is affected. This step, similar to SPLIT, is essential for determining whether the attributes are being used as shortcuts before implementing any bias mitigation techniques. However, the proposed technique allows us to evaluate the effectiveness of bias mitigation methods, such as GroupDRO or adversarial training, in removing shortcuts from this dataset. For example, we can first train a network with GroupDRO and then assess the extent to which it has removed shortcuts compared to the standard setup using the proposed method.

---

### Author Rebuttal · Authors · 2025-03-08

**Rebuttal:**

We thank the reviewers for their constructive feedback. We have carefully addressed their concerns and have uploaded a rebuttal PDF, highlighting all the edits made to the manuscript.

**Supporting Material:**

/attachment/190324872a8ae4fa9a8edbefc42093dc9855b120.pdf

---

### Comment · Area_Chair_fFVP · 2025-03-11

Dear authors, thank you for your rebuttal!

Dear reviewers, if you have not yet done so, could you please comment on the authors' answers and if needed adjust your score?

---

### Meta-Review · Area_Chair_fFVP · 2025-03-18

**Recommendation:** Accept (Poster)
**Confidence:** 4

**Metareview:**

The reviewers highlight the relevance and impact of the studied problem, due to increasing evidence of shortcuts and biases in medical imaging applications. The reviewers also overall agree on the quality of the writing, although request some clarifications, which the authors seem to have appropriately addressed in the rebuttal. There is a remaining question about experiments on counterfactual generation (reviewer hxHa) which could be addressed in future works of the authors.

As a personal note, the using statistical significance testing on k-fold cross-validation is not appropriate, because the different folds do not constitute fully independent samples. In general, it could be debated whether it is what we should be doing, see for example https://www.nature.com/articles/d41586-019-00857-9

Overall the paper seems of a good fit and quality for the MIDL conference so I would recommend accepting it.